# Dilated Recurrent Neural Networks

**Shiyu Chang**[1]*, **Yang Zhang**[1]*, **Wei Han**[2]*, **Mo Yu**[1], **Xiaoxiao Guo**[1], **Wei Tan**[1],
**Xiaodong Cui**[1], **Michael Witbrock**[1], **Mark Hasegawa-Johnson**[2], **Thomas S. Huang**[2]
[1]IBM Thomas J. Watson Research Center, Yorktown, NY 10598, USA
[2]University of Illinois at Urbana-Champaign, Urbana, IL 61801, USA
{shiyu.chang, yang.zhang2, xiaoxiao.guo}@ibm.com,
{yum, wtan, cuix, witbrock}@us.ibm.com,
{weihan3, jhasegaw, t-huang1}@illinois.edu

## Abstract

Learning with recurrent neural networks (RNNs) on long sequences is a notoriously difficult task. There are three major challenges: 1) complex dependencies, 2) vanishing and exploding gradients, and 3) efficient parallelization. In this paper, we introduce a simple yet effective RNN connection structure, the DILATEDRNN, which simultaneously tackles all of these challenges. The proposed architecture is characterized by multi-resolution **dilated recurrent skip connections**, and can be combined flexibly with diverse RNN cells. Moreover, the DILATEDRNN reduces the number of parameters needed and enhances training efficiency significantly, while matching state-of-the-art performance (even with standard RNN cells) in tasks involving very long-term dependencies. To provide a theory-based quantification of the architecture's advantages, we introduce a memory capacity measure, the **mean recurrent length**, which is more suitable for RNNs with long skip connections than existing measures. We rigorously prove the advantages of the DILATEDRNN over other recurrent neural architectures. The code for our method is publicly available[1].

## 1 Introduction

Recurrent neural networks (RNNs) have been shown to have remarkable performance on many sequential learning problems. However, long sequence learning with RNNs remains a challenging problem for the following reasons: first, memorizing extremely long-term dependencies while maintaining mid- and short-term memory is difficult; second, training RNNs using back-propagation-through-time is impeded by vanishing and exploding gradients; And lastly, both forward- and back-propagation are performed in a sequential manner, which makes the training time-consuming.

Many attempts have been made to overcome these difficulties using specialized neural structures, cells, and optimization techniques. Long short-term memory (LSTM) [10] and gated recurrent units (GRU) [6] powerfully model complex data dependencies. Recent attempts have focused on multi-timescale designs, including clockwork RNNs [12], phased LSTM [17], hierarchical multi-scale RNNs [5], *etc.* The problem of vanishing and exploding gradients is mitigated by LSTM and GRU memory gates; other partial solutions include gradient clipping [18], orthogonal and unitary weight optimization [2, 14, 24], and skip connections across multiple timestamps [8, 30]. For efficient sequential training, WaveNet [22] abandoned RNN structures, proposing instead the dilated causal convolutional neural network (CNN) architecture, which provides significant advantages in working directly with raw audio waveforms. However, the length of dependencies captured by a dilated CNN is limited by its kernel size, whereas an RNN's autoregressive modeling can, in theory, capture potentially infinitely

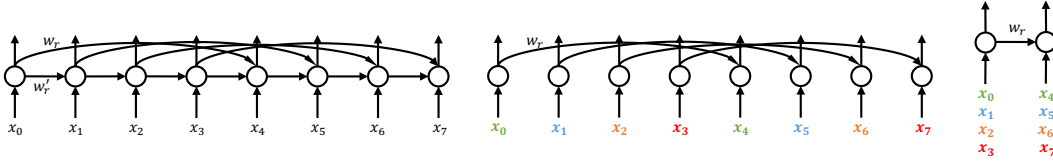

Figure 1: (left) A single-layer RNN with recurrent skip connections. (mid) A single-layer RNN with dilated recurrent skip connections. (right) A computation structure equivalent to the second graph, which reduces the sequence length by four times.

long dependencies with a small number of parameters. Recently, Yu *et al.* [27] proposed learning-based RNNs with the ability to jump (skim input text) after seeing a few timestamps worth of data; although the authors showed that the modified LSTM with jumping provides up to a six-fold speed increase, the efficiency gain is mainly in the testing phase.

In this paper, we introduce the DILATEDRNN, a neural connection architecture analogous to the dilated CNN [22, 28], but under a recurrent setting. Our approach provides a simple yet useful solution that tries to alleviate all challenges simultaneously. The DILATEDRNN is a multi-layer, and cell-independent architecture characterized by multi-resolution **dilated recurrent skip connections**. The main contributions of this work are as follows. 1) We introduce a new dilated recurrent skip connection as the key building block of the proposed architecture. These alleviate gradient problems and extend the range of temporal dependencies like conventional recurrent skip connections, but in the dilated version require fewer parameters and significantly enhance computational efficiency. 2) We stack multiple dilated recurrent layers with hierarchical dilations to construct a DILATEDRNN, which learns temporal dependencies of different scales at different layers. 3) We present the **mean recurrent length** as a new neural memory capacity measure that reveals the performance difference between the previously developed recurrent skip-connections and the dilated version. We also verify the optimality of the exponentially increasing dilation distribution used in the proposed architecture. It is worth mentioning that, the recent proposed Dilated LSTM [23] can be viewed as a special case of our model, which contains only one dilated recurrent layer with fixed dilation. The main purpose of their model is to reduce the temporal resolution on time-sensitive tasks. Thus, the Dilated LSTM is not a general solution for modeling at multiple temporal resolutions.

We empirically validate the DILATEDRNN in multiple RNN settings on a variety of sequential learning tasks, including long-term memorization, pixel-by-pixel classification of handwritten digits (with permutation and noise), character-level language modeling, and speaker identification with raw audio waveforms. The DILATEDRNN improves significantly on the performance of a regular RNN, LSTM, or GRU with far fewer parameters. Many studies [6, 14] have shown that vanilla RNN cells perform poorly in these learning tasks. However, within the proposed structure, even vanilla RNN cells outperform more sophisticated designs, and match the state-of-the-art. We believe that the DILATEDRNN provides a simple and generic approach to learning on very long sequences.

## 2 Dilated Recurrent Neural Networks

The main ingredients of the DILATEDRNN are its dilated recurrent skip connection and its use of exponentially increasing dilation; these will be discussed in the following two subsections respectively.

### 2.1 Dilated Recurrent Skip Connection

Denote $c_t^{(l)}$ as the cell in layer $l$ at time $t$. The dilated skip connection can be represented as

$$c_t^{(l)} = f\left(x_t^{(l)}, c_{t-s^{(l)}}^{(l)}\right). \tag{1}$$

This is similar to the regular skip connection[8, 30], which can be represented as

$$c_t^{(l)} = f\left(x_t^{(l)}, c_{t-1}^{(l)}, c_{t-s^{(l)}}^{(l)}\right). \tag{2}$$

$s^{(l)}$ is referred to as the skip length, or dilation of layer $l$; $x_t^{(l)}$ as the input to layer $l$ at time $t$; and $f(\cdot)$ denotes any RNN cell and output operations, *e.g.* Vanilla RNN cell, LSTM, GRU *etc.* Both skip connections allow information to travel along fewer edges. The difference between dilated and

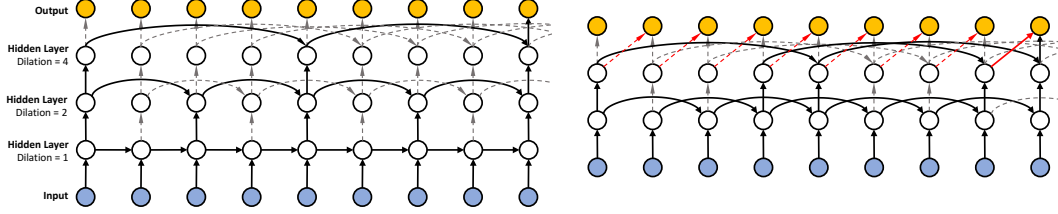

Figure 2: (left) An example of a three-layer DILATEDRNN with dilation 1, 2, and 4. (right) An example of a two-layer DILATEDRNN, with dilation 2 in the first layer. In such a case, extra embedding connections are required (red arrows) to compensate missing data dependencies.

regular skip connection is that the dependency on $c_{t-1}^{(l)}$ is removed in dilated skip connection. The left and middle graphs in figure 1 illustrate the differences between two architectures with dilation or skip length $s^{(l)} = 4$, where $W_r'$ is removed in the middle graph. This reduces the number of parameters.

More importantly, computational efficiency of a parallel implementation (*e.g.*, using GPUs) can be greatly improved by parallelizing operations that, in a regular RNN, would be impossible. The middle and right graphs in figure 1 illustrate the idea with $s^{(l)} = 4$ as an example. The input subsequences $\{x_{4t}^{(l)}\}$, $\{x_{4t+1}^{(l)}\}$, $\{x_{4t+2}^{(l)}\}$ and $\{x_{4t+3}^{(l)}\}$ are given four different colors. The four cell chains, $\{c_{4t}^{(l)}\}$, $\{c_{4t+1}^{(l)}\}$, $\{c_{4t+2}^{(l)}\}$ and $\{c_{4t+3}^{(l)}\}$, can be computed in parallel by feeding the four subsequences into a regular RNN, as shown in the right of figure 1. The output can then be obtained by interweaving among the four output chains. The degree of parallelization is increased by $s^{(l)}$ times.

## 2.2 Exponentially Increasing Dilation

To extract complex data dependencies, we stack dilated recurrent layers to construct DILATEDRNN. Similar to settings that were introduced in WaveNet [22], the dilation increases exponentially across layers. Denote $s^{(l)}$ as the dilation of the $l$-th layer. Then,

$$s^{(l)} = M^{l-1}, l = 1, \cdots, L. \tag{3}$$

The left side of figure 2 depicts an example of DILATEDRNN with $L = 3$ and $M = 2$. On one hand, stacking multiple dilated recurrent layers increases the model capacity. On the other hand, exponentially increasing dilation brings two benefits. First, it makes different layers focus on different temporal resolutions. Second, it reduces the average length of paths between nodes at different timestamps, which improves the ability of RNNs to extract long-term dependencies and prevents vanishing and exploding gradients. A formal proof of this statement will be given in section 3.

To improve overall computational efficiency, a generalization of our standard DILATEDRNN is also proposed. The dilation in the generalized DILATEDRNN does not start at one, but $M^{l_0}$. Formally,

$$s^{(l)} = M^{(l-1+l_0)}, l = 1, \cdots, L \text{ and } l_0 \geq 0, \tag{4}$$

where $M_0^l$ is called the starting dilation. To compensate for the missing dependencies shorter than $M^{l_0}$, a 1-by-$M^{(l_0)}$ convolutional layer is appended as the final layer. The right side of figure 2 illustrates an example of $l_0 = 1$, *i.e.* dilations start at two. Without the red edges, there would be no edges connecting nodes at odd and even time stamps. As discussed in section 2.1, the computational efficiency can be increased by $M^{l_0}$ by breaking the input sequence into $M^{l_0}$ downsampled subsequences, and feeding each into a $L - l_0$-layer regular DILATEDRNN with shared weights.

## 3 The Memory Capacity of DILATEDRNN

In this section, we extend the analysis framework in [30] to establish better measures of memory capacity and parameter efficiency, which will be discussed in the following two sections respectively.

### 3.1 Memory Capacity

To facilitate theoretical analysis, we apply the cyclic graph $\mathcal{G}_c$ notation introduced in [30].

**Definition 3.1** (Cyclic Graph). *The cyclic graph representation of an RNN structure is a directed multi-graph, $\mathcal{G}_C = (V_C, E_C)$. Each edge is labeled as $e = (u, v, \sigma) \in E_C$, where $u$ is the origin*

*node, $v$ is the destination node, and $\sigma$ is the number of time steps the edge travels. Each node is labeled as $v = (i, p) \in V_C$, where $i$ is the time index of the node modulo $m$, $m$ is the period of the graph, and $p$ is the node index. $\mathcal{G}_C$ must contain at least one directed cycle. Along the edges of any directed cycle, the summation of $\sigma$ must not be zero.*

Define $\mathscr{d}_i(n)$ as the length of the shortest path from any input node at time $i$ to any output node at time $i + n$. In [30], a measure of the memory capacity is proposed that essentially only looks at $\mathscr{d}_i(m)$, where $m$ is the period of the graph. This is reasonable when the period is small. However, when the period is large, the entire distribution of $\mathscr{d}_i(n), \forall n \leq m$ makes a difference, not just the one at span $m$. As a concrete example, suppose there is an RNN architecture of period $m = 10,000$, implemented using equation (2) with skip length $s^{(l)} = m$, so that $\mathscr{d}_i(n) = n$ for $n = 1, \cdots, 9,999$ and $\mathscr{d}_i(m) = 1$. This network rapidly memorizes the dependence on inputs at time $i$ of the outputs at time $i + m = i + 10,000$, but shorter dependencies $2 \leq n \leq 9,999$ are much harder to learn. Motivated by this, we proposed the following additional measure of memory capacity.

**Definition 3.2** (Mean Recurrent Length). *For an RNN with cycle $m$, the mean recurrent length is*

$$\bar{\mathscr{d}} = \frac{1}{m} \sum_{n=1}^{m} \max_{i \in V} \mathscr{d}_i(n). \tag{5}$$

Mean recurrent length studies the average dilation across different time spans within a cycle. An architecture with good memory capacity should generally have a small recurrent length for all time spans. Otherwise the network can only selectively memorize information at a few time spans. Also, we take the maximum over $i$, so as to punish networks that have good length only for a few starting times, which can only well memorize information originating from those specific times. The proposed mean recurrent length has an interesting reciprocal relation with the short-term memory (STM) measure proposed in [11], but mean recurrent length emphasizes more on long-term memory capacity, which is more suitable for our intended applications.

With this, we are ready to illustrate the memory advantage of DILATEDRNN . Consider two RNN architectures. One is the proposed DILATEDRNN structure with $d$ layers and $M = 2$ (equation (1)). The other is a regular $d$-layer RNN with skip connections (equation (2)). If the skip connections are of skip $s^{(l)} = 2^{l-1}$, then connections in the RNN are a strict superset of those in the DILATEDRNN , and the RNN accomplishes exactly the same $\bar{\mathscr{d}}$ as the DILATEDRNN , but with twice the number of trainable parameters (see section 3.2). Suppose one were to give every layer in the RNN the largest possible skip for any graph with a period of $m = 2^{d-1}$: set $s^{(l)} = 2^{d-1}$ in every layer, which is the regular skip RNN setting. This apparent advantage turns out to be a disadvantage, because time spans of $2 \leq n < m$ suffer from increased path lengths, and therefore

$$\bar{\mathscr{d}} = (m - 1)/2 + \log_2 m + 1/m + 1, \tag{6}$$

which grows linearly with $m$. On the other hand, for the proposed DILATEDRNN,

$$\bar{\mathscr{d}} = (3m - 1)/2m \log_2 m + 1/m + 1, \tag{7}$$

where $\bar{\mathscr{d}}$ only grows logarithmically with $m$, which is much smaller than that of regular skip RNN. It implies that the information in the past on average travels along much fewer edges, and thus undergoes far less attenuation. The derivation is given in appendix A in the supplementary materials.

## 3.2 Parameter Efficiency

The advantage of DILATEDRNN lies not only in the memory capacity but also the number of parameters that achieves such memory capacity. To quantify the analysis, the following measure is introduced.

**Definition 3.3** (Number of Recurrent Edges per Node). *Denote Card$\{\cdot\}$ as the set cardinality. For an RNN represented as $\mathcal{G}_C = (V_C, E_C)$, the number of recurrent edges per node, $N_r$, is defined as*

$$N_r = Card\left\{e = (u, v, \sigma) \in E_C : \sigma \neq 0\right\} / Card\{V_C\}. \tag{8}$$

Ideally, we would want a network that has large recurrent skips while maintaining a small number of recurrent weights. It is easy to show that $N_r$ for DILATEDRNN is 1 and that for RNNs with regular skip connections is 2. The DILATEDRNN has half the recurrent complexity as the RNN with regular skip RNN because of the removal of the direct recurrent edge. The following theorem states that the DILATEDRNN is able to achieve the best memory capacity among a class of connection structures with $N_r = 1$, and thus is among the most parameter efficient RNN architectures.

**Theorem 3.1** (Parameter Efficiency of DILATEDRNN). *Consider a subset of $d$-layer RNNs with period $m = M^{d-1}$ that consists purely of dilated skip connections (hence $N_r = 1$). For the RNNs in this subset, there are $d$ different dilations, $1 = s_1 \leq s_2 \leq \cdots \leq s_d = m$, and*

$$s_i = n_i s_{i-1}, \tag{9}$$

*where $n_i$ is any arbitrary positive integer. Among this subset, the $d$-layer DILATEDRNN with dilation rate $\{M^0, \cdots, M^{d-1}\}$ achieves the smallest $\bar{\mathscr{l}}$.*

The proof is motivated by [4], and is given in appendix B.

### 3.3 Comparing with Dilated CNN

Since DILATEDRNN is motivated by dilated CNN [22, 28], it is useful to compare their memory capacities. Although cyclic graph, mean recurrent length and number of recurrent edges per node are designed for recurrent structures, they happen to be applicable to dilated CNN as well. What's more, it can be easily shown that, compared to a DILATEDRNN with the same number of layers and dilation rate of each layer, a dilated CNN has exactly the same number of recurrent edges per node, and a slightly smaller (by $\log_2 m$) mean recurrent length. Hence both architectures have the same model complexity, and it looks like a dilated CNN has a slightly better memory capacity.

However, mean recurrent length only measures the memory capacity *within* a cycle. When going *beyond* a cycle, it is already shown that the recurrent length grows linearly with the number of cycles [30] for RNN structures, including DILATEDRNN, whereas for a dilated CNN, the receptive field size is always finite (thus mean recurrent length goes to infinity beyond the receptive field size). For example, with dilation rate $M = 2^{l-1}$ and $d$ layers $l = 1, \cdots, d$, a dilated CNN has a receptive field size of $2^d$, which is two cycles. On the other hand, the memory of a DILATEDRNN can go far beyond two cycles, particularly with the sophisticated units like GRU and LSTM. Hence the memory capacity advantage of DILATEDRNN over a dilated CNN is obvious.

## 4 Experiments

In this section, we evaluate the performance of DILATEDRNN on four different tasks, which include long-term memorization, pixel-by-pixel MNIST classification [15], character-level language modeling on the Penn Treebank [16], and speaker identification with raw waveforms on VCTK [26]. We also investigate the effect of dilation on performance and computational efficiency.

Unless specified otherwise, all the models are implemented with Tensorflow [1]. We use the default nonlinearities and RMSProp optimizer [21] with learning rate 0.001 and decay rate of 0.9. All weight matrices are initialized by the standard normal distribution. The batch size is set to 128. Furthermore, in all the experiments, we apply the sequence classification setting [25], where the output layer only adds at the end of the sequence. Results are reported for trained models that achieve the best validation loss. Unless stated otherwise, no tricks, such as gradient clipping [18], learning rate annealing, recurrent weight dropout [20], recurrent batch norm [20], layer norm [3], *etc.*, are applied. All the tasks are sequence level classification tasks, and therefore the "gridding" problem [29] is irrelevant. No "degridded" module is needed.

Three RNN cells, Vanilla, LSTM and GRU cells, are combined with the DILATEDRNN , which we refer to as dilated Vanilla, dilated LSTM and dilated GRU, respectively. The common baselines include single-layer RNNs (denoted as Vanilla RNN, LSTM, and GRU), multi-layer RNNs (denoted as stack Vanilla, stack LSTM, and stack GRU), and Vanilla RNN with regular skip connections (denoted as Skip Vanilla). Additional baselines will be specified in the corresponding subsections.

### 4.1 Copy memory problem

This task tests the ability of recurrent models in memorizing long-term information. We follow a similar setup in [2, 24, 10]. Each input sequence is of length $T + 20$. The first ten values are randomly generated from integers 0 to 7; the next $T - 1$ values are all 8; the last 11 values are all 9, where the first 9 signals that the model needs to start to output the first 10 values of the inputs. Different from the settings in [2, 24], the average cross-entropy loss is only measured at the last 10 timestamps. Therefore, the random guess yields an expected average cross entropy of $\ln(8) \approx 2.079$.

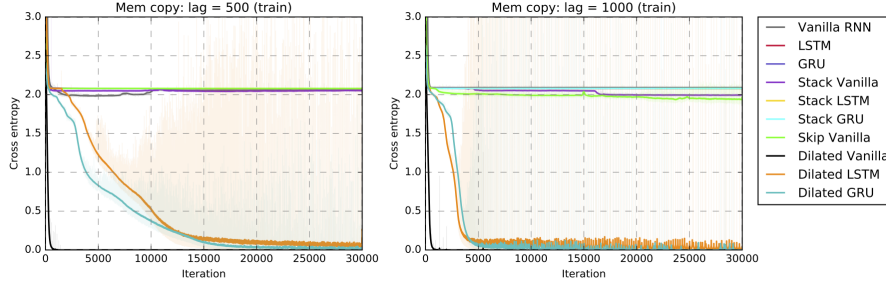

Figure 3: Results of the copy memory problem with $T = 500$ (left) and $T = 1000$ (right). The dilated-RNN converges quickly to the perfect solutions. Except for RNNs with dilated skip connections, all other methods are unable to improve over random guesses.

The DILATEDRNN uses 9 layers with hidden state size of 10. The dilation starts from one to 256 at the last hidden layer. The single-layer baselines have 256 hidden units. The multi-layer baselines use the same number of layers and hidden state size as the DILATEDRNN . The skip Vanilla has 9 layers, and the skip length at each layer is 256, which matches the maximum dilation of the DILATEDRNN.

The convergence curves in two settings, $T = 500$ and $1,000$, are shown in figure 3. In both settings, the DILATEDRNN with vanilla cells converges to a good optimum after about 1,000 training iterations, whereas dilated LSTM and GRU converge slower. It might be because the LSTM and GRU cells are much more complex than the vanilla unit. Except for the proposed models, all the other models are unable to do better than the random guess, including the skip Vanilla. These results suggest that the proposed structure as a simple renovation is very useful for problems requiring very long memory.

## 4.2 Pixel-by-pixel MNIST

Sequential classification on the MNIST digits [15] is commonly used to test the performance of RNNs. We first implement two settings. In the first setting, called the unpermuted setting, we follow the same setups in [2, 13, 14, 24, 30] by serializing each image into a 784 x 1 sequence. The second setting, called permuted setting, rearranges the input sequence with a fixed permutations. Training, validation and testing sets are the default ones in Tensorflow. Hyperparameters and results are reported in table 1. In addition to the baselines already described, we also implement the dilated CNN. However, the receptive fields size of a nine-layer dilated CNN is 512, and is insufficient to cover the sequence length of 784. Therefore, we added one more layer to the dilated CNN, which enlarges its receptive field size to 1,024. It also forms a slight advantage of dilated CNN over the DILATEDRNN structures.

In the unpermuted setting, the dilated GRU achieves the best evaluation accuracy of 99.2. However, the performance improvements of dilated GRU and LSTM over both the single- and multi-layer ones are marginal, which might be because the task is too simple. Further, we observe significant performance differences between stack Vanilla and skip vanilla, which is consistent with the findings in [30] that RNNs can better model long-term dependencies and achieves good results when recurrent skip connections added. Nevertheless, the dilated vanilla has yet another significant performance gain over the skip Vanilla, which is consistent with our argument in section 3, that the DILATEDRNN has a much more balanced memory over a wide range of time periods than RNNs with the regular skips. The performance of the dilated CNN is dominated by dilated LSTM and GRU, even when the latter have fewer parameters (in the 20 hidden units case) than the former (in the 50 hidden units case).

In the permuted setting, almost all performances are lower. However, the DILATEDRNN models maintain very high evaluation accuracies. In particular, dilated Vanilla outperforms the previous RNN-based state-of-the-art Zoneout [13] with a comparable number of parameters. It achieves test accuracy of 96.1 with only 44k parameters. Note that the previous state-of-the-art utilizes the recurrent batch normalization. The version without it has a much lower performance compared to all the dilated models. We believe the consistently high performance of the DILATEDRNN across different permutations is due to its hierarchical multi-resolution dilations. In addition, the dilated CNN is able the achieve the best performance, which is in accordance with our claim in section 3.3 that dilated CNN has a slightly shorter mean recurrent length than DILATEDRNN architectures, when sequence length fall within its receptive field size. However, note that this is achieved by adding one additional layer to expand its receptive field size compared to the RNN counterparts. When the useful information lies outside its receptive field, the dilated CNN might fail completely.

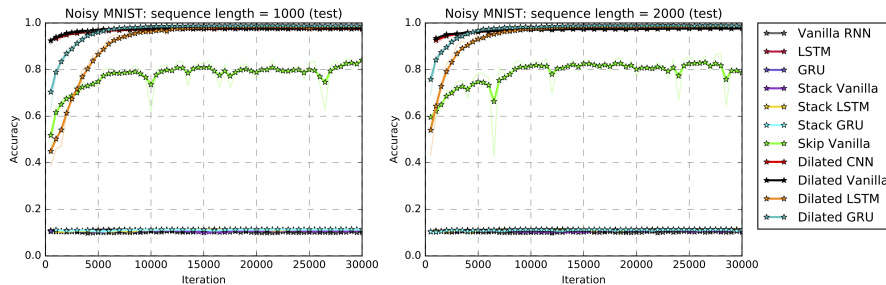

Figure 4: Results of the noisy MNIST task with $T = 1000$ (left) and 2000 (right). RNN models without skip connections fail. DILATEDRNN significant outperforms regular recurrent skips and on-pars with the dilated CNN.

Table 1: Results for unpermuted and permuted pixel-by-pixel MNIST. Italic numbers indicate the results copied from the original paper. The best results are bold.

| Method | # layers | hidden / layer | # parameters ($\approx$, k) | Max dilations | Unpermuted test accuracy | Permunted test accuracy |
|---|---|---|---|---|---|---|
| Vanilla RNN | 1 / 9 | 256 / 20 | 68 / 7 | 1 | - / 49.1 | 71.6 / 88.5 |
| LSTM [24] | 1 / 9 | 256 / 20 | 270 / 28 | 1 | *98.2* / 98.7 | 91.7 / 89.5 |
| GRU | 1 / 9 | 256 / 20 | 200 / 21 | 1 | 99.1 / 98.8 | 94.1 / 91.3 |
| IRNN [14] | 1 | 100 | 12 | 1 | *97.0* | *≈82.0* |
| Full uRNN [24] | 1 | 512 | 270 | 1 | *97.5* | 94.1 |
| Skipped RNN [30] | 1 / 9 | 95 / 20 | 16 / 11 | 21 / 256 | *98.1* / 85.4 | *94.0* / 91.8 |
| Zoneout [13] | 1 | 100 | 42 | 1 | - | *93.1* / *95.9*[2] |
| Dilated CNN [22] | 10 | 20 / 50 | 7 / 46 | 512 | 98.0 / 98.3 | 95.7 / **96.7** |
| Dilated Vanilla | 9 | 20 / 50 | 7 / 44 | 256 | 97.7 / 98.0 | 95.5 / 96.1 |
| Dilated LSTM | 9 | 20 / 50 | 28 / 173 | 256 | 98. 9 / 98.9 | 94.2 / 95.4 |
| Dilated GRU | 9 | 20 / 50 | 21 / 130 | 256 | 99.0 / **99.2** | 94.4 / 94.6 |

In addition to these two settings, we propose a more challenging task called the noisy MNIST, where we pad the unpermuted pixel sequences with $[0, 1]$ uniform random noise to the length of $T$. The results with two setups $T = 1,000$ and $T = 2,000$ are shown in figure 4. The dilated recurrent models and skip RNN have 9 layers and 20 hidden units per layer. The number of skips at each layer of skip RNN is 256. The dilated CNN has 10 layers and 11 layers for $T = 1,000$ and $T = 2,000$, respectively. This expands the receptive field size of the dilated CNN to the entire input sequence. The number of filters per layer is 20. It is worth mentioning that, in the case of $T = 2,000$, if we use a 10-layer dilated CNN instead, it will only produce random guesses. This is because the output node only sees the last $1,024$ input samples which do not contain any informative data. All the other reported models have the same hyperparameters as shown in the first three row of table 1. We found that none of the models without skip connections is able to learn. Although skip Vanilla remains learning, its performance drops compared to the first unpermuted setup. On the contrary, the DILATEDRNN and dilated CNN models obtain almost the same performances as before. It is also worth mentioning that in all three experiments, the DILATEDRNN models are able to achieve comparable results with an extremely small number of parameters.

## 4.3   Language modeling

We further investigate the task of predicting the next character on the Penn Treebank dataset [16]. We follow the data splitting rule with the sequence length of 100 that are commonly used in previous studies. This corpus contains 1 million words, which is small and prone to over-fitting. Therefore model regularization methods have been shown effective on the validation and test set performances. Unlike many existing approaches, we apply no regularization other than a dropout on the input layer. Instead, we focus on investigating the regularization effect of the dilated structure itself. Results are shown in table 2. Although Zoneout, LayerNorm HM-LSTM and HyperNetowrks outperform the DILATEDRNN models, they apply batch or layer normalizations as regularization. To the best of our knowledge, the dilated GRU with 1.27 BPC achieves the best result among models of similar sizes

Table 2: Character-level language modeling on the Penn Tree Bank dataset.

| Method | # layers | hidden / layer | # parameters ($\approx$, M) | Max dilations | Evaluation BPC |
|---|---|---|---|---|---|
| LSTM | 1 / 5 | 1k / 256 | 4.25 / 1.9 | 1 | 1.31 / 1.33 |
| GRU | 1 / 5 | 1k / 256 | 3.19 / 1.42 | 1 | 1.32 / 1.33 |
| Recurrent BN-LSTM [7] | 1 | 1k | - | 1 | *1.32* |
| Recurrent dropout LSTM [20] | 1 | 1k | 4.25 | 1 | *1.30* |
| Zoneout [13] | 1 | 1k | 4.25 | 1 | *1.27* |
| LayerNorm HM-LSTM [5] | 3 | 512 | - | 1 | *1.24* |
| HyperNetworks [9] | 1 / 2 | 1k | 4.91 / 14.41 | 1 | *1.26* / ***1.22***[3] |
| Dilated Vanilla | 5 | 256 | 0.6 | 64 | 1.37 |
| Dilated LSTM | 5 | 256 | 1.9 | 64 | 1.31 |
| Dilated GRU | 5 | 256 | 1.42 | 64 | 1.27 |

Table 3: Speaker identification on the VCTK dataset.

| | Method | # layers | hidden / layer | # parameters ($\approx$, k) | Min dilations | Max dilations | Evaluation accuracy |
|---|---|---|---|---|---|---|---|
| MFCC | GRU | 5 / 1 | 20 / 128 | 16 / 68 | 1 | 1 | 0.66 / **0.77** |
| Raw | Fused GRU | 1 | 256 | 225 | 32 / 8 | 32 /8 | 0.45 / 0.65 |
| | Dilated GRU | 6 / 8 | 50 | 103 / 133 | 32 / 8 | 1024 | 0.64 / 0.74 |

without layer normalizations. Also, the dilated models outperform their regular counterparts, Vanilla (didn't converge, omitted), LSTM and GRU, without increasing the model complexity.

## 4.4 Speaker identification from raw waveform

We also perform the speaker identification task using the corpus from VCTK [26]. Learning audio models directly from the raw waveform poses a difficult challenge for recurrent models because of the vastly long-term dependency. Recently the CLDNN family of models [19] managed to match or surpass the log mel-frequency features in several speech problems using waveform. However, CLDNNs coarsen the temporal granularity by pooling the first-layer CNN output before feeding it into the subsequent RNN layers, so as to solve the memory challenge. Instead, the DILATEDRNN directly works on the raw waveform without pooling, which is considered more difficult.

To achieve a feasible training time, we adopt the efficient generalization of the DILATEDRNN as proposed in equation (4) with $l_0 = 3$ and $l_0 = 5$ . As mentioned before, if the dilations do not start at one, the model is equivalent to multiple shared-weight networks, each working on partial inputs, and the predictions are made by fusing the information using a 1-by-$M^{l_0}$ convolutional layer. Our baseline GRU model follows the same setting with various resolutions (referred to as fused-GRU), with dilation starting at 8. This baseline has 8 share-weight GRU networks, and each subnetwork works on 1/8 of the subsampled sequences. The same fusion layer is used to obtain the final prediction. Since most other regular baselines failed to converge, we also implemented the MFCC-based models on the same task setting for reference. The 13-dimensional log-mel frequency features are computed with 25ms window and 5ms shift. The inputs of MFCC models are of length 100 to match the input duration in the waveform-based models. The MFCC feature has two natural advantages: 1) no information loss from operating on subsequences; 2) shorter sequence length. Nevertheless, our dilated models operating directly on the waveform still offer a competitive performance (Table 3).

## 4.5 Discussion

In this subsection, we first investigate the relationship between performance and the number of dilations. We compare the DILATEDRNN models with different numbers of layers on the noisy MNIST $T = 1,000$ task. All models use vanilla RNN cells with hidden state size 20. The number of dilations starts at one. In figure 5, we observe that the classification accuracy and rate of convergence increases as the models become deeper. Recall the maximum skip is exponential in the number of layers. Thus, the deeper model has a larger maximum skip and mean recurrent length.

Second, we consider maintaining a large maximum skip with a smaller number of layers, by increasing the dilation at the bottom layer of DILATEDRNN . First, we construct a nine-layer DILATEDRNN

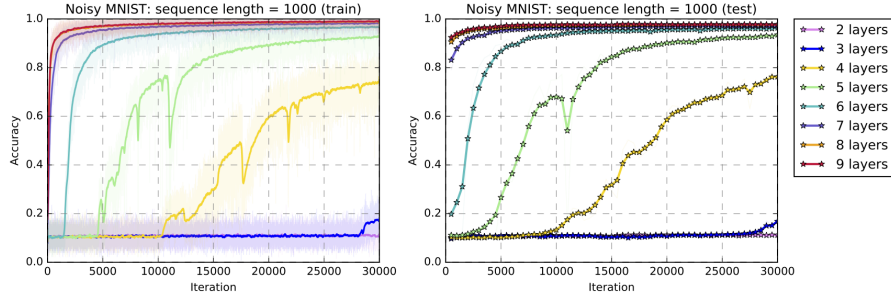

Figure 5: Results for dilated vanilla with different numbers of layers on the noisy MNIST dataset. The performance and convergent speed increase as the number of layers increases.

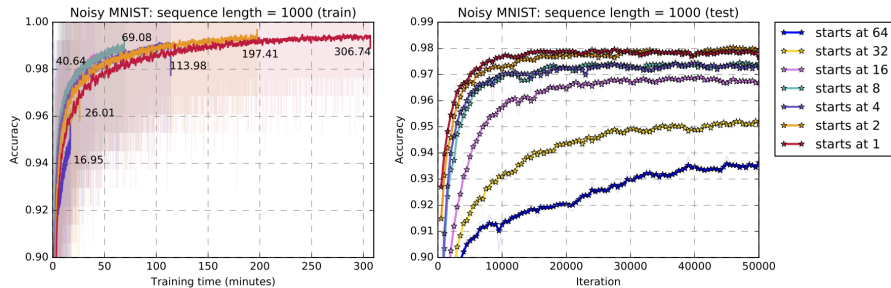

Figure 6: Training time (left) and evaluation performance (right) for dilated vanilla that starts at different numbers of dilations at the bottom layer. The maximum dilations for all models are 256.

model with vanilla RNN cells. The number of dilations starts at 1, and hidden state size is 20. This architecture is referred to as "starts at 1" in figure 6. Then, we remove the bottom hidden layers one-by-one to construct seven new models. The last created model has three layers, and the number of dilations starts at 64. Figure 6 demonstrates both the wall time and evaluation accuracy for 50,000 training iterations of noisy MNIST dataset. The training time reduces by roughly 50% for every dropped layer (for every doubling of the minimum dilation). Although the testing performance decreases when the dilation does not start at one, the effect is marginal with $s^{(0)} = 2$, and small with $4 \leq s^{(0)} \leq 16$. Notably, the model with dilation starting at 64 is able to train within 17 minutes by using a single Nvidia P-100 GPU while maintaining a 93.5% test accuracy.

# 5 Conclusion

Our experiments with DILATEDRNN provide strong evidence that this simple multi-timescale architectural choice can reliably improve the ability of recurrent models to learn long-term dependency in problems from different domains. We found that the DILATEDRNN trains faster, requires less hyperparameter tuning, and needs fewer parameters to achieve the state-of-the-arts. In complement to the experimental results, we have provided a theoretical analysis showing the advantages of DILATEDRNN and proved its optimality under a meaningful architectural measure of RNNs.

## Acknowledgement

Authors would like to thank Tom Le Paine (`paine1@illinois.edu`) and Ryan Musa (`ramusa@us.ibm.com`) for their insightful discussions.

## Footnotes

*Denotes equal contribution.

[1]https://github.com/code-terminator/DilatedRNN

[2]with recurrent batch norm [20].

[3]with layer normalization [3].

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
