[Supplementary Material · camera-ready-nips-supp.pdf]

**Supplementary Material: Dilated Recurrent Neural Networks**

## Appendix

## A    Mean Recurrent Length

This appendix gives the detailed derivation of the conclusions in section 3.1. Consider two RNN architectures. One is the proposed DILATEDRNN structure with $d$ layers. The other is a regular d-layer RNN with skip edges of length $2^{d-1}$ (hance $m = 2^{d-1}$), as shown in figure 1. For the regular skip RNN, it is obvious that $l_i(n)$ grows linearly within a cycle.

$$l_i(n) = \begin{cases} n+d & \text{if } n < m \\ 1+d & \text{if } n = m \end{cases},$$

and therefore

$$\bar{l} = \frac{1}{m}\left(\frac{m(m-1)}{2} + 1\right) + d = \frac{m-1}{2} + \log_2 m + \frac{1}{m} + 1,$$

which grows linearly with $m$. On the other hand, for the proposed DILATEDRNN structure, we have the following conclusion.

**Theorem A.1.** *For the* DILATEDRNN *with d layers.*

$$l_i(n) = \sum_{j=0}^{d-1} b_j + d, \forall n \leq m, \tag{10}$$

*where $b_0, \cdots, b_{\bar{j}}$ are digits of the binary representation of $n$, and $\bar{j}$ is the index of the highest binary bit. Thus*

$$\bar{l} = \frac{3m-1}{2m}\log_2 m + \frac{1}{m} + 1. \tag{11}$$

*Proof.* For any path that travels from input to output through $n$ time steps consists of edges that travel through time and those that travel through layers. Therefore

$$l_i(n) = r_i(n) + d, \tag{12}$$

where $r_i(n)$ is the minimum aggregate length of the edges that travel through time. $d$ is the minimum aggregate length of the edges that travel through layers, which is fixed. The problem of finding $l_i(n)$ is reduced to finding $r_i(n)$, which can then be reformulated as the change-making problem: Given a set of banknotes valued $\{2^0, 2^1, \cdots 2^{d-1}\}$ and an amount $n$. Denote the number of each banknote $\{a_1, \cdot, a_d$ make the amount, such that the total number of banknotes used is minimized. Formally

$$\min_{\{a_i\}} \sum_{j=1}^{d} a_j, \qquad \text{s.t.} \sum_{i=j}^{d} a_j 2^{j-1} = n. \tag{13}$$

Since dilations $s_i$'s are multiples of each other, the simple greedy algorithm suffices to find out the shortest path spanning $n$ time steps. That is, first use the largest skip edge, $s_d$, $\lfloor n/s_d \rfloor$ times, and then use the rest of the dilations to fit the residuals. This process is analogous to converting $n$ into its binary representation. Hence the optimal solution to equation (13), $\{a_i^*\}$ is given by

$$a_i^* = b_i. \tag{14}$$

For $n$ traversing 1 through $m$, each $a_i^*$ will be 1 50% of the time, except for $a_{d-1}^*$, which equals one only once. Therefore

$$\bar{l} = \frac{1}{m}\left(\frac{m-1}{2}(d-1) + 1\right) + d = \frac{3m-1}{2m}\log_2 m + \frac{1}{m} + 1. \tag{15}$$

$\square$

# B Optimality of the Proposed Skip Distribution

This appendix provides the proof to theorem 3.1. By analogy to the change-making problem, this theorem can be reformulated as the optimal denomination problem, which involves finding the a set of banknote denominations $1 = s_1 \leq s_2 \leq \cdots \leq s_L = m$ such that the average number of banknotes for making the change of values ranging from 1 to $m$, i.e. $\bar{d}$, is minimized.

The optimal denomination problem remains to be an open problem in mathematics, but solutions are readily available when the candidate denominations are confined to those satisfying equation (9), as shown in [4]. The proof here is adapted from that in [4].

**Proof to theorem 3.1:**

*Proof.* First, it is easy to show that the RNN architecture that minimizes $\bar{d}$ must have the same dilation rate within the same layer, because 1) it has all the paths that consist of all the combinations of recurrent edges with different dilations, where the optimal shortest paths must lie; 2) in such architectures $d_i(n)$ does not depend on $i$, so that the maximum over $i$ in equation (5) does not have an effect.

Now that we have confined the candidate set, the problem is reduced to finding a set of $1 = s_1 \leq s_2 \leq \cdots \leq s_L = m$ such that $\bar{d}$ is minimized. We can apply equation (12),

$$r_i(n) = \operatorname*{argmin}_{\{a_i\}} \sum_{j=1}^{d} a_i, \qquad \text{s.t.} \sum_j a_j s_j = n. \tag{16}$$

Define

$$\bar{r} = \frac{1}{m} \sum_{n=1}^{m} \max_i r_i(n), \tag{17}$$

as the average number of recurrent edge usage. Hence minimizing $\bar{d}$ is further reduced to minimizing $\bar{r}$. Since dilations $s_i$'s are multiples of each other, the simple greedy algorithm suffices to find out the shortest path spanning $n$ time steps. That is, first use the largest skip edge, $s_d$, $\lfloor n/s_d \rfloor$ times, and then use the rest of the skip lengths to fit the residuals. Therefore, to fit all the time spans ranging from 0 to $m - 1$, the histogram of uses of recurrent edge of length $s_i < m$ per time span is distributed uniformly across 0 through $s_{i+1}/s_i$ time. Formally, the total uses of recurrent skips of length $s_i < m$ is

$$\frac{m}{2}(\frac{s_{i+1}}{s_i} - 1). \tag{18}$$

To fit the time span $m$, only the edge $s_d = m$ will be used once. Hence

$$\bar{r} = \frac{1}{2}(\sum_{i=1}^{d-1} \frac{s_{i+1}}{s_i} - d + 1). \tag{19}$$

Since the arithmetic mean is alway greater than or equal to the geometric mean

$$\sum_{i=1}^{d-1} \frac{s_{i+1}}{s_i} \geq \sqrt[d-1]{m} = M, \tag{20}$$

with equality if and only if

$$\frac{s_{i+1}}{s_i} = M, \forall i. \tag{21}$$

$\square$