[Reviews · NeurIPS 2017]

Reviewer 1



The paper is relevant to the community, well-written and decently structured. I have no major concerns, only some interested questions: - Why do the other networks not learn at all in some of the tasks? / What is the deeper reason for this? - How does the proposed memory capacity measure relate to the memory capacity introduced by Jaeger in the context of Echo State Networks? Minor comments: -p.3: subseuqnces -> subsequences -p.6: "We implement two setting." -> settings -p.6: "which is consistent with our argument in 3" -> in Section 3 -p.8: "Figure 6 demonstrates both the wall time" -> overall time?

Reviewer 2



The paper proposes to build RNNs out of recurrent layers with skip connections of exponentially increasing lenghts, i.e. the recurrent equivalent of stacks of dilated convolutions. Compared to previous work on RNNs with skip connections, the connections to the immediately preceding timestep are fully replaced here, allowing for increased parallelism in the computations. Dilated RNNs are compared against several other recurrent architectures designed to improve learning of long-range temporal correlations, and found to perform comparably or better for some tasks. The idea is simple and potentially effective, and related work is adequately covered in the introduction, but the evaluation is somewhat flawed. While the idea of using dilated connections in RNNs is clearly inspired by their use in causal CNNs (e.g. WaveNet), the evaluation never includes them. Only several other recurrent architectures are compared against. I think this is especially limiting as the paper claims computational advantages for dilated RNNs, but these computational advantages could be even more outspoken for causal CNNs, which allow for training and certain forms of inference to be fully parallelised across time. The noisy MNIST task described from L210 onwards (in Section 4.2) seems to be nonstandard and I don't really understand the point. It's nice that dilated RNNs are robust to this type of corruption, but sequences padded with uniform noise seem fairly unlikely to occur in the real world. In Section 4.3, raw audio modelling is found to be less effective than MFCC-based modelling, so this also defeats the point of the dilated RNN a bit. Also, trying dilated RNNs combined with batch normalization would be an interesting additional experiment. Overall, I don't find the evidence quite as strong as it is claimed to be in the conclusion of the paper. Remarks: - Please have the manuscript proofread for spelling and grammar. - In the introduction, it is stated that RNNs can potentially model infinitely long dependencies, but this is not actually achievable in practice (at least with the types of models that are currently popular). To date, this advantage of RNNs over CNNs is entirely theoretical. - It is unclear whether dilated RNNs also have repeated stacks of dilation stages, like e.g. WaveNet. If not, this would mean that large-scale features (from layers with large dilation) never feed into fine-scale features (layers with small dilation). That seems like it could be a severe limitation, so this should be clarified. - Figures 4, 5, 6 are a bit small and hard to read, removing the stars from the curves might make things more clear. UPDATE: in the author feedback it is stated that the purpose of the paper is to propose an enhancement for RNN models, but since this enhancement is clearly inspired by dilation in CNNs, I don't think this justifies the lack of a comparison to dilated CNNs at all. Ultimately, we want to have models that perform well for some given task -- we do not want to use RNNs just for the sake of it. That said, the additional experiments that were conducted do address my concerns somewhat (although I wish they had been done on a different task, I still don't really get the point of "noisy MNIST".) So I have raised the score accordingly.

Reviewer 3



The paper studies from a theoretical point of view the benefits of using dilated skip connections for RNN to improve their long-term memorization capabilities. A new neural memory measure - mean recurrent length - is introduced that shows that the dilated setting is superior to the skip connection setting, while having less parameters. Interestingly, the dilated setting allows for parallelisation leading to lower computational time. Experiments on the copy problem and mnist show significantly better performance compared to the vanilla counterparts, while experiments on language modelling (Penn tree bank) and speaker identification yield competitive results, but with simpler settings. Comments: As the authors point out, a dilated LSTM setup was already proposed in Vezhnevets et al, Feudal networks. It is unclear in what way their "fundamental basis and target tasks are very different"; as in this paper, the goal there is to improve long term memorization. Instead, it would be beneficial to discuss the results obtained here in light of their findings. E.g. they found that dilated LSTM on its own does not perform better than vanilla LSTM. Is the stacked dilated model that makes the results in this paper outperform the vanilla versions in all cases? In Dilated residual networks, by Yu et al, the authors discuss the potential gridding artefacts from using exponentially increasing dilations in stacked models. It would be useful to discuss such issues in the recurrent setup as well. Few typos: line 48 temporal; line 78 subsequences;